# Methionine Restriction Improves Cognitive Ability by Alleviating Hippocampal Neuronal Apoptosis through H19 in Middle-Aged Insulin-Resistant Mice

**DOI:** 10.3390/nu14214503

**Published:** 2022-10-26

**Authors:** Chuanxing Feng, Yuge Jiang, Shiying Li, Yueting Ge, Yonghui Shi, Xue Tang, Guowei Le

**Affiliations:** 1State Key Laboratory of Food Science and Technology, Jiangnan University, Wuxi 214122, China; 2Center for Food Nutrition and Functional Food Engineering, School of Food Science and Technology, Jiangnan University, Wuxi 214122, China; 3Key Laboratory of Neuroregeneration of Jiangsu and Ministry of Education, Co-Innovation Center of Neuroregeneration, Nantong University, Nantong 226019, China; 4College of Life Science, Xinyang Normal University, Xinyang 464000, China

**Keywords:** methionine restriction, neuron apoptosis, cognitive ability, insulin resistance, oxidative stress

## Abstract

LncRNA H19 has been reported to regulate apoptosis and neurological diseases. Hippocampal neuron apoptosis damages cognitive ability. Methionine restriction (MR) can improve cognitive impairment. However, the effect of MR on hippocampal neuronal apoptosis induced by a high-fat diet (HFD) in middle-aged mice remains unclear. For 25 weeks, middle-aged mice (C57BL/6J) were given a control diet (CON, 0.86% methionine + 4.2% fat), a high-fat diet (HFD, 0.86% methionine + 24% fat), or an HFD + MR diet (HFMR, 0.17% methionine + 24% fat). The HT22 cells were used to establish the early apoptosis model induced by high glucose (HG). In vitro, the results showed that MR significantly improved cell viability, suppressed the generation of ROS, and rescued HT22 cell apoptosis in a gradient-dependent manner. In Vivo, MR inhibited the damage and apoptosis of hippocampal neurons caused by a high-fat diet, reduced hippocampal oxidative stress, improved hippocampal glucose metabolism, relieved insulin resistance, and enhanced cognitive ability. Furthermore, MR could inhibit the overexpression of H19 and caspase-3 induced by HFD, HG, or H_2_O_2_ in vivo and in vitro, and promoted let-7a, b, e expression. These results indicate that MR can protect neurons from HFD-, HG-, or H_2_O_2_-induced injury and apoptosis by inhibiting H19.

## 1. Introduction

There is a strong relationship between obesity and cognitive impairment in midlife. HFD-induced metabolic disturbances may promote brain aging and cognitive decline [1]. Long-term HFD has been shown to cause global and hippocampal insulin resistance [2], and this increases with age [3]. MR can extend the lifespan [4], restore a youthful metabolic phenotype in adult mice [5], enhance insulin sensitivity [6], and promote glucose metabolism [7]. Insulin resistance with hyperglycemia and defective insulin signaling results in neurons lacking energy and becoming vulnerable to oxidative damage and apoptosis [8,9,10], accelerates cognitive decline. Studies have shown that the first manifestation of Alzheimer’s disease (AD) is neuronal death, not extracellular amyloid plaques [11]. Therefore, interventions targeting neuronal apoptosis may help to reduce the cognitive risk associated with obesity and aging.

Patients with type 2 diabetes also have cognitive impairment, which is related to the severity of hyperglycemia [12], due to imbalances in glucose metabolism in the brain, particularly in the hippocampus [13]. A significant decrease in glucose metabolism was also observed in AD and amnestic mild cognitive impairment, which may be partly due to oxidative damage of enzymes in glycolysis, tricarboxylic acid cycle, and ATP biosynthesis [14].

H19 is a long non-coding RNA (LncRNA) that does not translate into protein, but can interact with miRNA as a competitive endogenous RNA and participate in the regulation of target gene expression [15]. And it is involved in the regulation of apoptosis and neurological diseases [16,17,18,19]. The expression of H19 is increased in brain injuries induced by AD [20], diabetes [21], ischemia/reperfusion [22], epilepsy [23], and oxygen and glucose deprivation [24]. Furthermore, high expression of H19 is related to neuronal apoptosis. Overexpression of H19 exacerbates status epilepticus (SE)-induced neuronal apoptosis in the hippocampus, whereas inhibition of H19 protects rats from SE-induced cellular damage [25]. In a cellular model of AD, β-amyloid protein increased H19 expression and inhibited cell viability in PC12 cells, while silencing of H19 promoted cell viability and inhibited apoptosis in PC12 cells induced by Aβ_25–35_ [20]. LncRNA can sponge the microRNAs (miRNAs) and suppress its inhibitory effects [15]. MiRNAs are small non-coding RNAs that regulate gene expression at the post-transcriptional level [26]. Several studies have shown that miRNAs have a role in neurodegenerative disease [27], and are involved in many biological processes such as developmental timing, differentiation, and cell death [28]. The let-7 family is a series of miRNAs that participate in the regulation of apoptosis by targeting caspase-3 mRNA [29,30,31,32].

Previous studies have revealed the effects of methionine restriction (MR) on cognitive deficits at different ages by ameliorating hippocampal inflammation and oxidative stress [33,34,35]. However, the protective effect of MR on hippocampal neuronal apoptosis in middle-aged insulin-resistant mice and its potential mechanisms remain unclear. Methionine is a DNA methyl donor, and MR plays an epigenetic role [36]. By regulating S-adenosylhomocysteine hydrolase, H19 modifies genome-wide DNA methylation [37]. MR can regulate oxidative stress [38,39], whereas H19 expression responds to changes in the level of oxidative stress [40]. Therefore, MR may have some functional association with H19. Recent research has indicated that methionine deficiency can regulate the expression of lncRNA pvt1 in gastric cancer cells [41], indicating that MR has the potential to regulate lncRNA expression.

In this study, we evaluated the effect of H19 on MR in improving HFD-induced hippocampal apoptosis, and we analyzed the interrelation of hippocampal H19, oxidative stress, and miRNAs.

## 2. Materials and Methods

### 2.1. Animal Experiment

All studies were approved by the Jiangnan University Animal Care and Use Ethics Committee (JN. No20190530c0840215) and conducted under the National Guidelines for Experimental Animal Welfare. C57BL/6J male mice (8 months old, approximately 32 g) (SCXK (Su) 2018-0008) were purchased from GemPharmatech Co., Ltd. (Nanjing, China) and reared with free water and food in a specific pathogen free (SPF) level laboratory at the Animal Experiment Center of Jiangnan University (Wuxi, China). The 24 mice were randomly and equally divided into three groups (CON, HFD, and HFMR) according to their body weight after 1 week. For 25 weeks, mice were given either a control diet (CON, 0.86% methionine + 4.2% fat), a high-fat diet (HFD, 0.86% methionine + 24% fat), or an HFD + MR diet (HFMR, 0.17% methionine + 24% fat). The formulae for the diets are listed in Appendix A. The experiment lasted for 25 weeks, and the body weights of the mice were measured weekly.

### 2.2. Tests of Novel Object Recognition and Morris Water Maze

In this study, nonspatial recognition memory was assessed using a Novel Object Recognition (NOR) test according to the previously described protocol [42]. The Morris Water Maze (MWM) test was used to assess spatial recognition memory according to a previously described protocol [43].

### 2.3. TdT-Mediated dUTP Nick End Labeling (TUNEL) Fluorescent Assay and Nissl Staining Assay

Hippocampal apoptosis was detected by TUNEL fluorescent staining using the One- Step TUNEL Apoptosis Assay Kit according to the manufacturer’s instructions (Beyotime, Shanghai, China). DAPI staining was used for nuclei staining. The staining was observed using fluorescence microscopy in a 200× visual field (Olympus Corporation, Tokyo, Japan).

For histological analysis, hippocampal neurons were observed by Nissl staining according to the specification of the Nissl Staining Solution (Beyotime, Shanghai, China). The sections were observed using a microscope in 100×, 200×, and 400× visual fields (Olympus Corporation, Tokyo, Japan).

### 2.4. Detection of Oxidative Stress Index, ROS Levels, Fasting Glucose, and Fasting Insulin

The levels of malondialdehyde (MDA), superoxide dismutase (SOD), and total antioxidant capacity (T-AOC) in hippocampus were detected using assay kits (Nanjing Jiancheng Bioengineering Institute, Nanjing, China). Reactive oxygen species (ROS) were tested by the luminol-dependent chemiluminescence method according to the previously described protocol [44]. Fasting glucose was measured using OneTouch Ultra Test Strips (Johnson Medical Equipment and Materials Company Limited, Shanghai, China). Fasting insulin was measured using an ELISA kit (Huijia, Guangzhou, China). Homeostasis model assessment of insulin resistance (HOMA-IR) index = [fasting glucose levels (mmol/L)] × [fasting insulin (mIU/L)]/22.5 [45].

### 2.5. Cell Culture and Transfection

Mouse hippocampal neuron HT22 cells were purchased from the Chinese Academy of Sciences cell bank (Shanghai, China). HT22 cells were cultured in high-glucose basic DMEM (Gbico, Carlsbad, CA, USA) with 10% FBS (Gbico, Carlsbad, CA, USA) and 1% penicillin-streptomycin solution (10,000 U/mL penicillin and 10 mg/mL streptomycin) (Gbico, Carlsbad, CA, USA). MR (0%, 20%, 40%, 60%, 80%, 90%, 95%, 98%, 100%) was achieved by incubating the HT22 cells in methionine-free DMEM (Gbico, Carlsbad, CA, USA) supplemented with 10% FBS (3.38 mg/L methionine and 1.88 mg/L L-cystine), 1% penicillin-streptomycin solution, L-methionine (27, 21.6, 16.2, 10.8, 5.4, 2.7, 1.35, 0.54, 0 mg/L; Sigma, Louis, MO, USA), L-cystine·2HCl (56, 44.8, 33.6, 22.4, 11.2, 5.6, 2.8, 1.12, 0 mg/L; Sigma, Louis, MO, USA), and L-glutamine (520 mg/L; Sigma, Louis, MO, USA) for up to 12 h. H19-expressing plasmid pEX-3-H19 and pEX-3 empty vectors were designed and sold by GenePharma Co., Ltd. (Suzhou, China). Transfection was conducted using Lipo8000 (Beyotime, Shanghai, China), according to the manufacturer’s protocol. After 48 h of transfection, the transfected cells were intervened with HG or MR.

### 2.6. Cell Viability Assay

HT22 cell viability was tested using the cell counting kit-8 (CCK-8) assay kit (Nanjing Jiancheng Bioengineering Institute, Nanjing, China). HT22 cells were cultured in a 96-well plate (5 × 10^3^ cells/well) for 24 h and then cultured in MR (0−100%) or HG (225 mM) culture medium for 12 h. Then, CCK-8 solution (10 μL/well) was added and incubated for 1.5 h in the CO_2_ incubator. Absorbance was measured using a microplate reader at 450 nm (Epoch, Biotek, Winooski, VT, USA). Cell viability (%) = [A(treatment) − A(blank)]/[A(control) − A(blank)].

### 2.7. Measurement of ROS in HT22

ROS levels were measured using 2,7-dichlorodihydrofluorescein diacetate (DCFH-DA), which is a specific ROS proprietary fluorogenic probe. After plate incubation, the prepared DCFH-DA probe (10 μM) was added to each well and incubated for 20 min. The intensity of fluorescence was detected using a fluorescent microplate reader at 488 nm excitation and 525 nm emission (Synergy H4, Biotek, Winooski, VT, USA). The radio of ROS in samples was normalized to the CON group [46]. ROS fluorescence images were photographed using the FITC channel of a wide-field imaging high-connotation system (Image Xpress Micro XLS, Molecular Devices, Sunnyvale, CA, USA).

### 2.8. Flow Cytometric Examination

The annexin V/PI double staining method was used to quantify the levels of apoptosis in each group. HT22 cells were seeded in six-well plates (5 × 10^5^ cells/well) and incubated for 24 h (37 °C, 5% CO_2_), and then treated with 225 mM HG with or without different levels of MR for 12 h. After 12 h of culture, the supernatant and adherent cells were collected. A total of 2 × 10^5^ cells were taken and resuspended with 500 µL of binding solution, and then stained with 5 µL of FITC and PI for 10 min at room temperature, followed by detection of the apoptotic cells using flow cytometry (FACSAriaII, Becton, Dickinson and Company, Franklin Lakes, NJ, USA) with 1 × 10^4^ cells in each group.

### 2.9. Quantitative Real-Time Polymerase Chain Reaction (qRT-PCR)

Total RNA was purified from the hippocampus of mice and HT22 cells using total RNA extraction reagent (Vazyme, Nanjing, China) according to the manufacturer’s protocol. Total cDNA was synthesized using the HiScript III RT SuperMix for qPCR (+gDNA wiper) reagent kit (Vazyme, Nanjing, China). The Mir-X miRNA first strand was synthesized with the Mir-X miRNA First-Strand Synthesis Kit (Takara Bio USA, Mountain View, CA, USA). qRT-PCR was performed using an AceQ qPCR SYBR Green Master Mix (High ROX Premixed) kit (Vazyme, Nanjing, China) on a Quantagene q225 Real-Time PCR system (kubo Technology Ltd., Beijing, China) according to the manufacturer’s protocol. The relative expression of mRNA or miRNA in each sample was normalized to β-actin (mRNA) or U6 (miRNA) by using the 2^−ΔΔCt^ method. The primer sequences are listed in Appendix A.

### 2.10. Western Blot Analysis

The proteins were extracted from the hippocampus of mice and HT22 cells using RIPA lysis buffer (Beyotime, Shanghai, China). The concentration of protein was determined using a BCA kit (Beyotime, Shanghai, China). The protein was separated by SDS- PAGE, followed by transfer onto PVDF membranes. Subsequently, the membranes were incubated with primary antibodies specific for caspase-3 or β-actin (1:1000, Abcam). Then the membranes were incubated with secondary antibodies (1:10,000, Abcam), and bands were visualized with an ECL Chemiluminescent Substrate Reagent kit (Beyotime, Shanghai, China). Image J software was used to analyze the bands, and the expression ratios were normalized to β-actin.

### 2.11. Statistical Analysis

All data were analyzed using IBM SPSS 20.0 software. The data were expressed as the mean ± standard error of the mean (SEM). The difference between groups was evaluated using one-way ANOVA followed by a Dunnett’s multiple comparisons test. A *p*-value of <0.05 indicated a significant difference.

## 3. Results

### 3.1. MR Improved Cognitive Ability in HFD-Induced Insulin-Resistant Mice

Many studies have shown that a long-term HFD can cause cognitive decline [47]. The NOR and MWM tests were used to examine how MR affected the cognitive ability of middle-aged mice fed with HFD.

The nonspatial recognition memory ability of mice was assessed by the time difference between exploring old (OA) and new objects (OB) in the NOR (Figure 1A). During the test, as shown in Figure 1D, mice in the CON group and HFMR group spent significantly more time exploring OB than OA (*p* < 0.05), while it was not significantly different in the HFD group (*p* > 0.05). The tracking paths (Figure 1B) showed that the mice in the CON and HFMR groups rather than the HFD group tended more toward OB. Furthermore, as shown in Figure 1E, by calculating the discrimination index (DI), it can be seen that the DI of the HFD group was significantly lower than that of the CON and HFMR groups (*p* < 0.05).

The MWM (Figure 1F) is a classic test of spatial recognition memory. As shown in Figure 1H, the HFD group mice took significantly more time to find the platform after the fourth day compared with the CON group (*p* < 0.05), and mice in the HFMR group effectively reduced the time to find the platform (*p* < 0.05). Meanwhile, in the probe trial (Figure 1G,J,K), HFMR group mice spent significantly more time in the target quadrant and crossed the platform more frequently than HFD group mice (*p* < 0.05), and the motion paths of the CON and HFMR group mice were mainly distributed in the platform quadrant, while the HFD group mice moved without a target. Therefore, MR can restore the cognitive ability of the middle-aged mice fed with HFD to be similar to that of the CON group.

### 3.2. MR Improved Insulin Resistance and Insulin Signaling in HFD Mice

There is sufficient evidence that HFD could lead to systemic insulin resistance, including suppression of insulin signaling in the hippocampus [48]. In this study, HFD significantly increased the levels of fasting glucose, fasting insulin, and HOMA-IR compared to the CON group (*p* < 0.05, Table 1). Moreover, HFD mice showed insulin resistance symptoms. The MR diet significantly reduced the levels of fasting glucose, fasting insulin and HOMA-IR (*p* < 0.05) compared to HFD insulin-resistant mice (Table 1).

Glucose serves as the exclusive fuel source for energy production in the brain [49]. Hippocampal glucose metabolism is important for cognitive function, and impaired glucose metabolism in the hippocampus is a risk factor for cognitive impairment [50]. As shown in Figure 2, HFD reduced the glucose metabolism-related mRNA expression of insulin receptor substrate 1 (IRS-1), glucose transporter 1 (GLUT1), hexokinase 2 (HK2) in the hippocampus, compared to the CON group (*p* < 0.05). However, these mRNA expressions were significantly increased in MR-treated HFD mice. But, the mRNA expression of M2 pyruvate kinase (PKM2) and phosphofructokinase 1 (PFK1) did not present significant differences in the HFD and HFMR groups in comparison to the CON group (*p* > 0.05).

### 3.3. MR Improved Hippocampal Oxidative Stress and Inhibited H19 Expression in HFD-Induced Insulin-Resistant Mice

Impaired insulin signaling makes neurons more susceptible to oxidative stress [8]. As shown in Figure 3, HFD significantly increased the levels of MDA and ROS, inhibited the T-AOC and SOD activities and reduced the mRNA expression of NADPH quinineoxidoreductase-1 (NQO-1) and heme oxygenase-1 (HO-1) (*p* < 0.05). As expected, MR can restore the levels of oxidative stress and ROS to similar to those found in the CON group. Compared with HFD group, MR significantly deceased MDA and ROS levels, increased SOD activity and T-AOC, and promoted the mRNA expression of nuclear factor erythroid-2-related factor 2 (Nrf2), NQO-1 and HO-1 (*p* < 0.05). Meanwhile, HFD induced H19 overexpression in the hippocampus, whereas MR significantly reduced H19 expression in the HFD insulin-resistant mice (*p* < 0.05). In addition, there was a strong positive correlation between H19 expression and ROS levels (Figure 3F).

### 3.4. MR Alleviated Neuronal Injury and Apoptosis in Hippocampus of HFD-Induced Insulin-Resistant Mice

Increased expression of H19 is related to neuronal apoptosis [25]. The results of TUNEL staining showed that a chronic HFD led to significant apoptosis of hippocampal neurons in insulin-resistant mice compared with the CON group (*p* < 0.01, Figure 4A). HFD-induced apoptosis in the hippocampal was significantly inhibited by MR, which reduced proapoptotic genes (Bcl-2 associated X protein, Bax and cysteinyl aspartate specific proteinase-3, and caspase-3) expression and promoted antiapoptotic genes (B-cell lymphoma 2, Bcl-2) expression (*p* < 0.05). In addition, MR significantly decreased the protein expression of caspase-3 (*p* < 0.05).

Photos of the Nissl-stained mice hippocampus are shown in Appendix A. In the CON group, the morphology of hippocampus CA3 neurons was normal, the cytoplasm was rich in Nissl bodies, and the nucleus was clearly visible. In the HFD group, degeneration of and reduction in neurons, nuclear atrophy, and reduction in Nissl bodies were observed. However, MR protected neurons from damage induced by HFD, and it reduced the damaged hippocampal neurons. In this study, the mRNA expression of several neurotrophic factors was investigated to explain the neuroprotective effect of MR against HFD. As shown in Appendix A, HFD significantly downregulated the mRNA expression of brain-derived neurotrophic factor (BDNF), tyrosine kinase receptor B (TRκB), Ca^2+^/calmodulin-dependent protein kinase II alpha chain (CAMK2A), synaptopodin (Synpo), and cAMP response element-binding protein (CREB) compared with the CON group (*p* < 0.05). However, the MR diet intervention upregulated these mRNA expressions compared with the HFD group (*p* < 0.05).

### 3.5. MR Inhibited the High Expression of H19 Induced by High Glucose in HT22 Cells

To explore the effect of MR on H19, HT22 cells were treated with different MR levels and 225 mM glucose for 12 h. The results of the CCK-8 experiment (Figure 5A) showed that the activity of HT22 cells increased with increasing MR levels when the MR level was less than 80%, but activity decreased when the restriction degree was greater than 80%. Cell viability was strongest at MR80%, and MR inhibited the decrease of cell viability caused by HG (Figure 5B).

As shown in Figure 5, HG increased ROS content (Figure 5D) and H19 expression (Figure 5C) in a concentration-dependent manner, which was inhibited by MR treatment.

### 3.6. MR Inhibited the High Expression of H19 Induced by ROS in HT22 Cells

Elevated expression of H19 was observed in HFD-fed mice and HT22 cells cultured with different concentrations of HG. Since HFD or HG can cause oxidative stress and ROS production, we suspect that H19 expression increases in response to ROS in hippocampal neurons.

H_2_O_2_ is an important ROS [51]. Therefore, to explore the effect of ROS on H19, different concentrations of H_2_O_2_ were used to intervene cells for 1 h. As shown in Figure 6A,B, H_2_O_2_ increased ROS content and H19 expression in a concentration-dependent manner, with a Pearson correlation coefficient between H19 and ROS of 0.8634 (Figure 6C). MR could significantly inhibit the increase of ROS and restore it to similar to those found in the CON group. In addition, MR significantly inhibited H19 overexpression induced by 0.75 mM H_2_O_2_ (*p* < 0.05, Figure 6D,E,G), and the Pearson correlation coefficient between H19 and ROS was 0.7166 (Figure 6F). These results support our hypothesis that ROS production is related to H19 overexpression in hippocampal neurons.

### 3.7. MR Inhibited HT22 Cell Apoptosis through Inhibiting H19

Flow cytometry was used to quantify the level of apoptosis. As shown in Figure 7E,G, HG (225mM) significantly increased the apoptosis of HT22 cell compared to the CON group (*p* < 0.01). Compared with the HG group, MR significantly inhibited apoptosis in a gradient-dependent manner even at high glucose concentrations with apoptosis rates of about 60% (*p* < 0.01).

MR improved the upregulation of Bax and caspase-3 mRNAs, as well as the downregulation of Bcl-2 mRNA induced by HG in a gradient-dependent manner. A Pearson correlation analysis indicated that there was a positive correlation of H19 with apoptosis, caspase-3, and Bax, but a negative correlation with Bcl-2. Additionally, compared to Bax and Bcl-2, caspase-3, H19, and apoptosis had a greater positive correlation coefficient.

We further investigated the relationship among MR, H19, caspase-3, and apoptosis. The results showed that the mRNA expression of caspase-3 and Bax was significantly upregulated (Figure 8A), the protein expression of caspase-3 (Figure 8B) was significantly upregulated, and the mRNA expression of Bcl-2 was significantly downregulated after H19 was overexpressed (*p* < 0.05, Figure 8A). It was previously demonstrated that H19 could sponge the let-7 family, and let-7 (let-7 a, b, c, e, f) was specifically targeted to reduce caspase-3 expression [29,30,31,32]. Our results indicated that MR significantly promoted the expression of let-7a-5p, let-7b-5p, and let-7e-5p (Figure 8 F,G), which in turn significantly inhibited the expression of caspase-3 and apoptosis (*p* < 0.05). In contrast, the opposite result appeared when H19 was overexpressed (Figure 8F). Thus, MR likely inhibited apoptosis in HT22 cells through the H19/let-7/caspase-3 pathway.

## 4. Discussion

This study found that MR can ameliorate apoptosis of hippocampal neurons and improve cognitive ability. In vitro, we found that MR inhibited ROS generation and apoptosis of HT22 cells induced by HG. This is the first report on the inhibitory effect of MR on apoptosis of hippocampal neurons.

Long-term HFD can cause oxidative stress in the hippocampus and systemic insulin resistance [2,52]. Systemic insulin resistance induces hyperglycemia, and excess glucose causes neurotoxicity by increasing apoptosis and inhibiting cell proliferation, with cognitive impairment [9,10]. Patients with type 2 diabetes also have cognitive impairment [12]. Oxidative stress is an important cause of hippocampal neuron damage, as well as the decline in learning and memory ability [47]. It is because oxidative stress leads to an increase in ROS levels in the hippocampus [52], which leads to neuronal damage and apoptosis [53]. In our results, 25 weeks of HFD increased oxidative stress and ROS levels in the hippocampus, resulted in global insulin resistance, elevated blood glucose, and decreased cognitive ability of mice. There was increased neuronal apoptosis and damaged neurons in the hippocampal CA3 region. In mice fed HFD, MR reduced oxidative stress and ROS levels, inhibited hippocampal neurons apoptosis, improved cognitive ability, improved insulin resistance, decreased blood glucose levels, and restored these too similar to the CON group. At the transcription level, MR increased the expression of glucose metabolism-related mRNA (IRS-1, GLUT1, and HK2) and the expression of learning and memory-related mRNA (BDNF, TRκB, CAMK2A, Synpo, and CREB) in HFD mice.

H19 is one of the first LncRNAs to be identified [54]. It is highly expressed in fetal tissues and can promote embryonic development and growth [55], but its expression is significantly reduced after birth [56]. However, H19 is reactivated in some pathological states, including tissue regeneration, cancer, hypoxia, and CNS disease [57,58]. H19 overexpression significantly increases Bax and caspase-3 expression but decreases Bcl-2 expression, thereby promoting the apoptosis of hippocampal neuronal cells [21]. H19 upregulation induces apoptosis during oxygen and glucose deprivation and reperfusion (OGD/R) via autophagic hyperactivation [59]. Our experimental results in vivo and in vitro showed that H19 was overexpressed in the hippocampus of mice fed HFD and HT22 cells intervened by HG or H_2_O_2_. In vitro, HG significantly reduced the viability and increased the apoptosis of HT22 cells. The same findings were observed in vivo, where HFD resulted in hippocampal neuronal apoptosis. Because LncRNA participates in gene regulation at the transcriptional level [60], we detected the expression of apoptosis related mRNA. In vivo, HFD decreased Bcl-2 mRNA expression, increased Bax mRNA expression, and increased caspase-3 mRNA and protein expression. In vitro, HG inhibited Bcl-2 mRNA expression, and promoted Bax and caspase-3 mRNA expression. Moreover, Pearson correlation analysis showed that there was a strong positive correlation between H19 and apoptosis and caspase-3 mRNA expression. Thus, H19 overexpression is associated with hippocampal neuronal apoptosis.

H19 expression responds to changes in oxidative stress [40]. Several studies have shown that H19 increases when cells are exposed to ROS. For example, in cholangiocarcinoma cell lines (QBC939, sk-cha-1, and RBE) and glioma cell lines (U251 and ln229), both short-term and long-term oxidative stress induced by H_2_O_2_ or glucose upregulated the expression of H19 in cells [61,62]. In nucleus pulposus cells, an H_2_O_2_-induced increase in H19 expression led to cell aging and proliferation suppression, and it stagnated the cell cycle in the G0/G1 phase [46]. Furthermore, H19 overexpression induced by H_2_O_2_ was enhanced with increasing ROS levels, and H19 expression could be regulated by modulating ROS levels [63]. In this study, oxidative stress induced by HFD promoted ROS production and H19 expression in the hippocampus, accompanied by neuronal apoptosis. In Vitro, H19 expression in HT22 cells increased with increasing concentrations of H_2_O_2_ or glucose and was positively correlated with the ROS levels. However, MR alleviated oxidative stress in mice fed with HFD by promoting SOD activity and T-AOC, reducing MDA and ROS levels, and increasing the mRNA expression of Nrf2, NQO-1, and HO-1, as well as inducing a significant decrease in H19 expression. In vitro, MR was also found to decrease H_2_O_2_ or HG-induced ROS accumulation and H19 expression in a gradient-dependent manner. The results showed that HFD or HG promoted the expression of H19 by promoting ROS production. And MR likely regulated H19 expression by ameliorating ROS production.

Neuronal apoptosis was ameliorated after knockdown of H19 in brain injury [20]. In Aβ_25–35_-induced AD and other brain injuries induced by glucose/oxygen deprivation or hypoxia injury, H19 silencing inhibits the expression of Bax and caspase-3, as well as promotes Bcl-2 expression, thereby inhibiting apoptosis [20,22,24]. In our study, MR inhibited H19 expression by reducing ROS production, ameliorated neuronal apoptosis, inhibited the mRNA and protein expression of caspase-3, reduced Bax mRNA expression, and promoted Bcl-2 mRNA expression in the hippocampus of HFD mice. In vitro studies also revealed that MR could inhibit HG-induced H19 overexpression and apoptosis of HT22 cells and promote cell viability. In addition, MR arranged the neurons orderly and alleviated the morphological abnormalities of neurons in the hippocampal CA3 area of HFD mice. However, after H19 overexpression, the inhibitory effect of MR on apoptosis of HT22 cells was diminished, the mRNA and protein expression of caspase-3 and Bax mRNA expression were increased, and the mRNA expression of Bcl-2 was decreased. Thus, H19 likely mediated the regulatory effects of MR on hippocampal neuronal apoptosis. Although studies have shown that H19 plays an essential function in neuronal regulation and MR has a significant effect on regulating energy metabolism and improving learning and memory ability, this is a new discovery, and no one has previously studied the effect of MR on H19 expression before.

In adult newborn neurons, the let-7 family is abundantly expressed and is implicated in the control of neuronal migration [64]. H19 can regulate the availability of let-7 by acting as a molecular sponge [65]. Some studies have shown that the let-7 family (let-7a, b, c, e, f) binds to the 3′-UTR of caspase-3, inhibits caspase-3 mRNA translation, then inhibits caspase-3 protein expression [29,30,31,32]. At the early stage of AD, neurons show apoptosis, changes in miRNA expression, and caspase-3 expression increases [66]. These studies showed that the let-7 family and caspase-3 were involved in the regulation of neuronal apoptosis and were related to learning and memory ability. In this study, HFD inhibited the expression of let-7a, b, c, e, f, while MR significantly promoted the expression of let-7a, b, e. In vitro, the overexpression of H19 weakened the promotion effect of MR on let-7, along with an increase in caspase-3 expression and apoptosis. Therefore, MR likely inhibited the expression of H19, reduced the combination of H19 and let-7, and increased the expression of let-7a, b, e. The combination of let-7 and caspase-3 reduced the expression of caspase-3 mRNA and protein and inhibited cell apoptosis.

## 5. Conclusions

In summary, MR can alleviate oxidative stress, and inhibit oxidative stress-induced H19 overexpression in the hippocampus. Then, MR reduces hippocampal neuronal apoptosis through the H19/let-7/caspase-3 pathway (Figure 9), thereby improving the cognitive ability of middle-aged mice fed with HFD. This finding can provide a dietary reference for preventing or alleviating cognitive impairment in the elderly.

## Figures and Tables

**Figure 1 nutrients-14-04503-f001:**
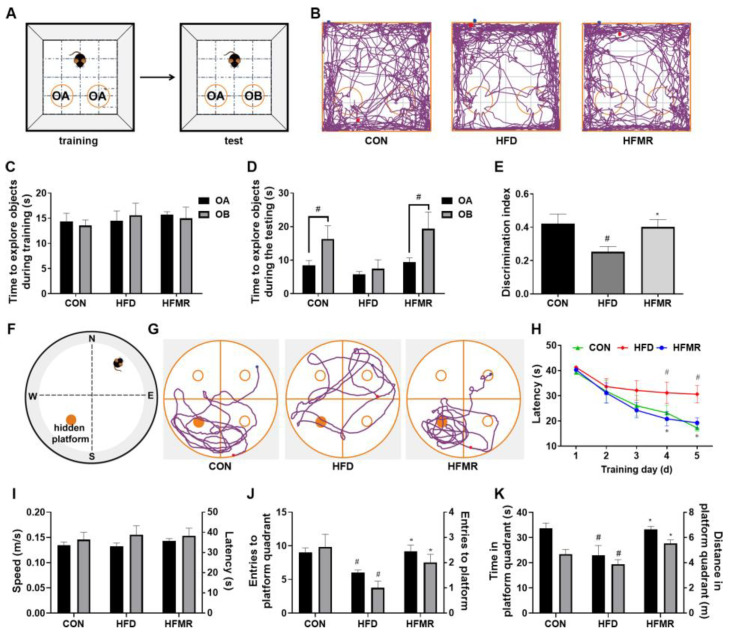
Effects of MR on cognitive ability in HFD-induced insulin-resistant mice. (**A**) Diagram of the NOR. (**B**) Tracking paths of mice in NOR. (**C**) Time to explore objects during training. (**D**) Time to explore objects during the test. (**E**) Discrimination index (DI). (**F**) Diagram of the MWM. (**G**) Tracking paths of mice in MWM. (**H**) Latency in the platform hiding period. (**I**) Speed and latency in the platform visibility period. (**J**) Number of entries into the platform quadrant and the platform in the platform hiding period. (**K**) Staying time and moving distance in the platform quadrant during the platform hiding period. All data are shown as the mean ± SEM (*n* = 8). ^#^
*p* < 0.05 (HFD vs. CON); * *p* < 0.05 (HFMR vs. HFD).

**Figure 2 nutrients-14-04503-f002:**
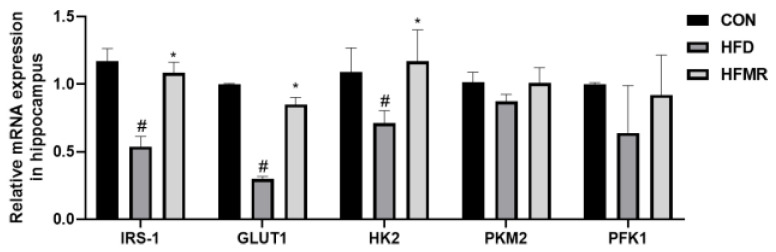
Effects of MR on insulin signaling in the hippocampus of HFD-induced obesity mice. All data are shown as the mean ± SEM (*n* = 8). ^#^
*p* < 0.05, (HFD vs. CON); * *p* < 0.05 (HFMR vs. HFD).

**Figure 3 nutrients-14-04503-f003:**
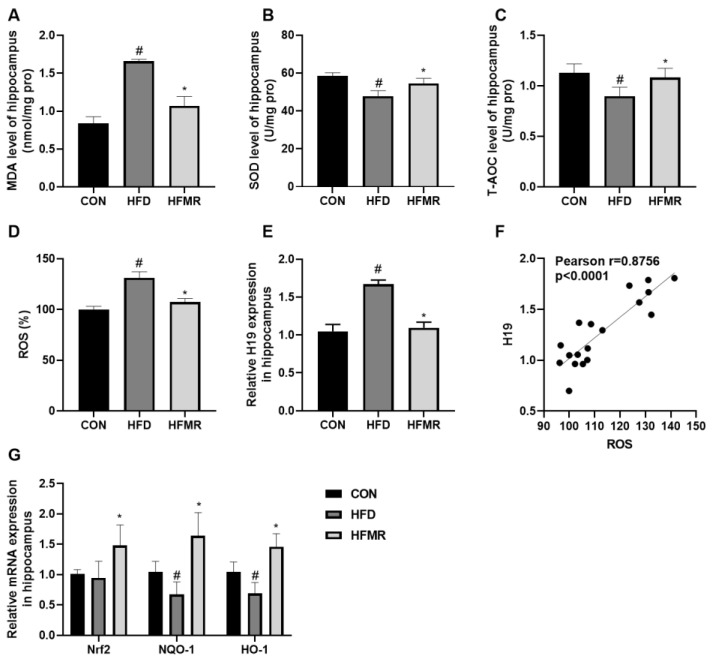
Effects of MR on hippocampal oxidative stress and H19 expression in HFD-induced insulin-resistant mice. (**A**) MDA level of the hippocampus. (**B**) SOD level of the hippocampus. (**C**) T-AOC level of the hippocampus. (**D**) Relative ROS level of the hippocampus. (**E**) Relative H19 expression in the hippocampus. (**F**) Pearson correlation coefficient between ROS and H19. (**G**) Nrf2 signaling pathway-related mRNA expression. All data are shown as the mean ± SEM (*n* = 8). ^#^
*p* < 0.05 (HFD vs. CON); * *p* < 0.05 (HFMR vs. HFD).

**Figure 4 nutrients-14-04503-f004:**
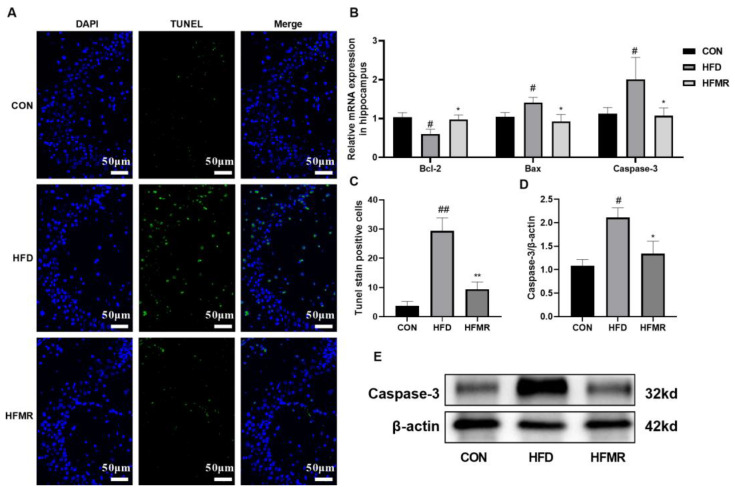
Effects of MR on neuronal apoptosis in the hippocampus of HFD-induced insulin-resistant mice. (**A**) TUNEL fluorescent assay of the hippocampus (200×). (**B**) mRNA expression of apoptosis-related genes. (**C**) The number of TUNEL-positive cells. (**D**) Relative protein expression of caspase-3 in hippocampus. (**E**) Caspase-3 western blotting bands. All data are shown as the mean ± SEM (*n* = 8). ^#^
*p* < 0.05, ^##^
*p* < 0.01 (HFD vs. CON); * *p* < 0.05, ** *p* < 0.01 (HFMR vs. HFD).

**Figure 5 nutrients-14-04503-f005:**
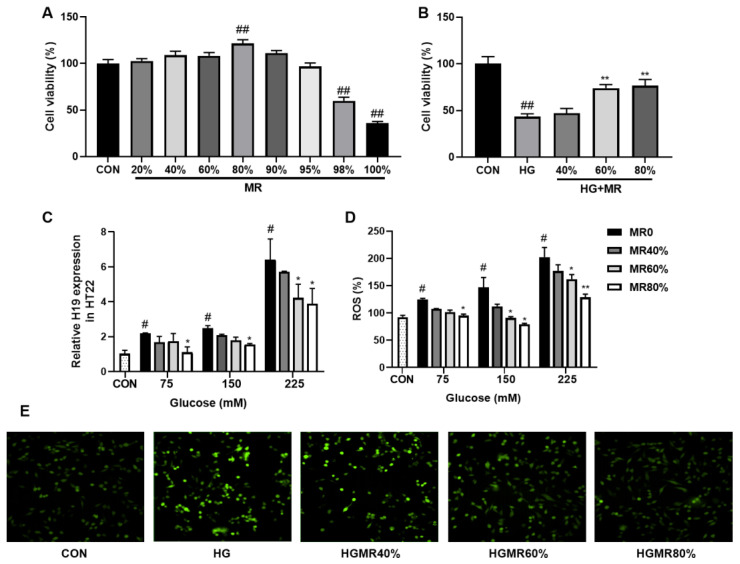
Effects of MR on the expression of H19 induced by high glucose in HT22 cells. (**A**) HT22 cell viability treated with different levels of MR. (**B**) HT22 cell viability treated with HG (225 mM) and different levels of MR. (**C**) Relative H19 expression of HT22 cells treated with different concentrations of glucose and different levels of MR. (**D**) Relative ROS levels of HT22 cells treated with different concentrations of glucose and different levels of MR. (**E**) DCFH fluorescence in HT22 cells treated with HG (225 mM) and different levels of MR. All data are shown as the mean ± SEM. ^#^
*p* < 0.05, ^##^
*p* < 0.01 (vs. CON); * *p* < 0.05, ** *p* < 0.01 (vs. HG).

**Figure 6 nutrients-14-04503-f006:**
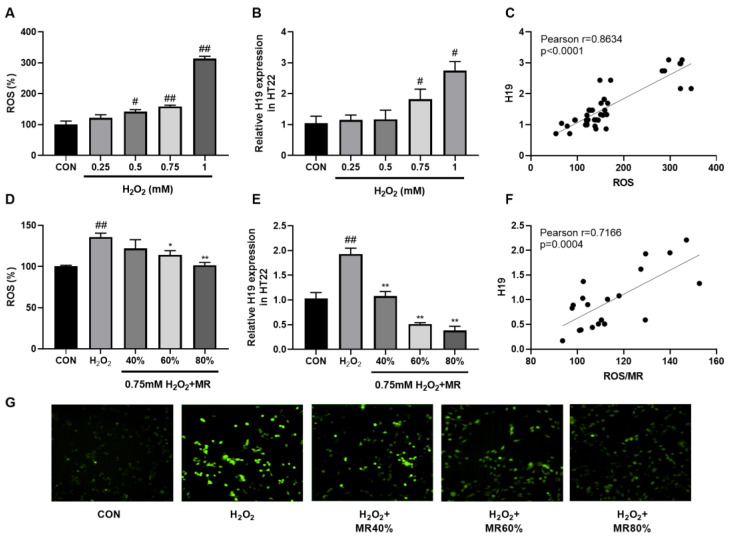
Effects of MR on the high expression of H19 induced by ROS in HT22 cells. (**A**) Relative ROS levels induced by different concentrations of H_2_O_2_ in HT22 cells. (**B**) Relative H19 expression in HT22 cells treated with different concentrations of H_2_O_2_. (**C**) Pearson correlation between H19 and ROS in HT22 cells treated with different concentrations of H_2_O_2_. (**D**) Relative ROS levels in HT22 cells treated with H_2_O_2_ (0.75 mM) and different levels of MR. (**E**) Relative expression of H19 in HT22 cells treated with H_2_O_2_ (0.75 mM) and different levels of MR. (**F**) Pearson correlation between H19 and ROS in HT22 cells treated with H_2_O_2_ (0.75 mM) and different levels of MR. (**G**) DCFH fluorescence in HT22 cells treated with H_2_O_2_ (0.75 mM) and different levels of MR. All data are shown as the mean ± SEM. ^#^
*p* < 0.05, ^##^
*p* < 0.01 (vs. CON); * *p* < 0.05, ** *p* < 0.01 (vs. H_2_O_2_).

**Figure 7 nutrients-14-04503-f007:**
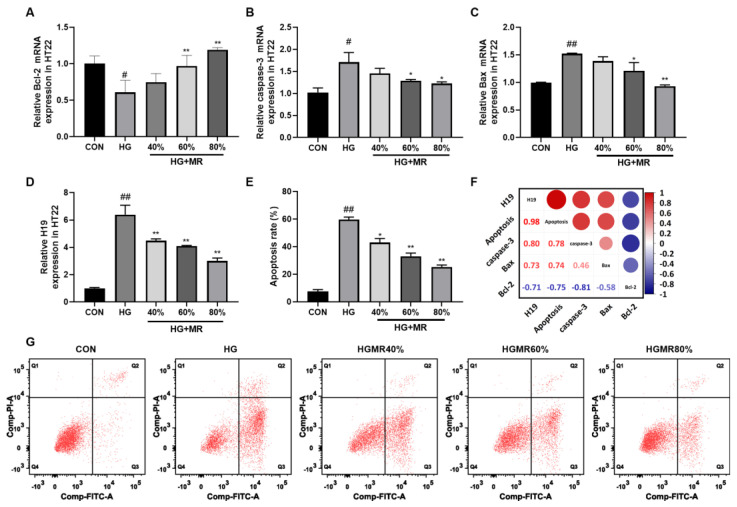
Effects of MR on HT22 cell apoptosis induced by HG. (**A**–**D**) Relative Bcl-2, caspase-3, Bax mRNA, and H19 expression in HT22 cells treated with HG (225 mM) and different levels of MR. (**E**,**G**) The apoptosis of HT22 cells treated with HG (225 mM) and different levels of MR is presented by flow cytometry. (**F**) Pearson’s correlations among H19, apoptosis, caspase-3, Bcl-2, and Bax. The size and color of the circles in the upper right corner of the matrix indicate the correlation index level (red denoting a positive correlation and blue denoting a negative correlation), while the bottom left corner of the matrix contains the associated correlation index’s numerical value. All data are shown as the mean ± SEM. ^#^
*p* < 0.05, ^##^
*p* < 0.01 (vs. CON); * *p* < 0.05, ** *p* < 0.01 (vs. HG).

**Figure 8 nutrients-14-04503-f008:**
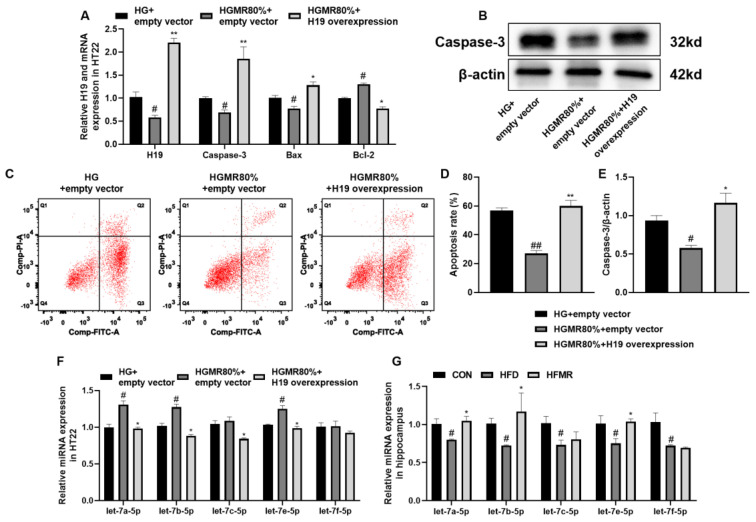
MR inhibited HT22 cell apoptosis likely through H19/let-7/caspase-3 pathway. (**A**) Relative H19 and mRNA expression in HT22. (**B**) Caspase-3 western blotting bands. (**C**,**D**) The apoptosis of HT22 cells. (**E**) Relative protein expression of caspase-3 in HT22. (**F**) Relative miRNA expression in HT22. (**G**) Relative miRNA expression in hippocampus. All data are shown as the mean ± SEM. ^#^
*p* < 0.05, ^##^
*p* < 0.01 (vs. CON or HG + empty vector); * *p* < 0.05, ** *p* < 0.01 (vs. HFD or HGMR80% + empty vector).

**Figure 9 nutrients-14-04503-f009:**
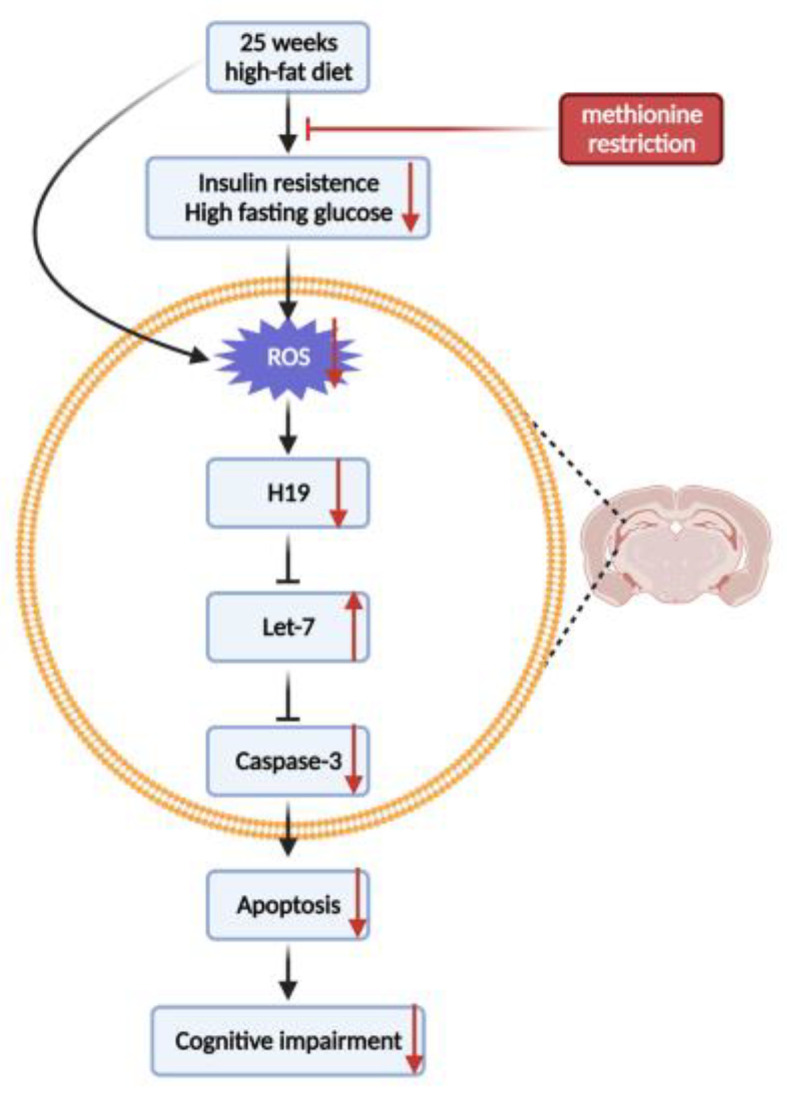
The mechanism of MR on improving cognitive ability by reducing hippocampal neuronal apoptosis in middle-aged insulin-resistant mice.

**Table 1 nutrients-14-04503-t001:** Effects of MR on insulin resistance of HFD-induced obese mice.

	CON	HFD	HFMR
Body weight (g)	41.63 ± 1.33	53.03 ± 0.31 ^##^	41.28 ± 3.54 **
Fasting glucose (mmol/L)	9.56 ± 0.20	11.22 ± 0.35 ^##^	9.11 ± 0.26 **
Fasting insulin (mIU/L)	7.52 ± 0.20	8.43 ± 0.30 ^#^	7.05 ± 0.24 *
HOMA-IR	3.20 ± 0.26	4.21 ± 0.41 ^#^	2.86 ± 0.23 *

All data are shown as the mean ± SEM (*n* = 8). ^#^
*p* < 0.05, ^##^
*p* < 0.01 (HFD vs. CON); * *p* < 0.05, ** *p* < 0.01 (HFMR vs. HFD).

## Data Availability

All data are included in the articles and Appendix A.

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
