# Peer review of "Methionine Restriction Improves Cognitive Ability by Alleviating Hippocampal Neuronal Apoptosis through H19 in Middle-Aged Insulin-Resistant Mice"

_nutrients, 2022, doi:10.3390/nu14214503_

Round 1
Reviewer 1 Report
The authors of this manuscript conducted a study of the role of long non-coding H19 RNA, as well as the role of methionine restriction in the apoptosis mechanism of hippocampal neurons in standard experimental models in vitro and in vivo. The content of the work corresponds to the subject of the journal, the tasks and goal set by the authors. The work was done at a fairly high level and the data obtained are of interest. However, I did not find table S1 and figure S2 in the manuscript, to which the authors refer. In general, I can recommend this manuscript for publication after it has been supplemented with missing material and minor corrections.
Minor comments.
1. It is necessary to supplement the manuscript with table S1 and figure S2.
2. The introduction should briefly describe the role of the let-7 family in apoptosis.
3. The authors should describe in more detail how the methionine restriction was carried out.
4. Expand abbreviations (lncRNA, SPF, HOMA-IR).
5. L.56: “LncRNA can sponge the miRNA” -> “LncRNA can sponge the microRNAs (miRNA)”
Author Response
Dear professor,
Thank you for your valuable comments on this paper. I have revised the manuscript according to your suggestions. Thank you for reviewing my paper again. If there are any other suggestions, please let me know and I will revise them again.
Details of the revisions are as follows:
Minor comments:
- It is necessary to supplement the manuscript with table S1 and figure S2.
I have put the supplementary materials at the end of the manuscript.
- The introduction should briefly describe the role of the let-7 family in apoptosis.
I have described the role of the let-7 family in the introduction according to your advice.
Line 62-63: “The let-7 family is a series of miRNAs that participate in the regulation of apoptosis by targeting Caspase-3 [29-32].”
- The authors should describe in more detail how the methionine restriction was carried out.
I have described the method of methionine restriction in the section 2.5 according to your advice.
Line 122-127: “MR (0%, 20%, 40%, 60%, 80%, 90%, 95%, 98%, 100%) was achieved by incubating the HT22 cells in methionine-free DMEM (Gbico, California, USA) supplemented with 10% FBS (3.38 mg/L methionine and 1.88 mg/L L-cystine), 1% penicillin-streptomycin solution, L-methionine (27, 21.6, 16.2, 10.8, 5.4, 2.7, 1.35, 0.54, 0 mg/L; Sigma, Louis, MO, USA), L-cystine·2HCl (56, 44.8, 33.6, 22.4, 11.2, 5.6, 2.8, 1.12, 0 mg/L; Sigma, Louis, MO, USA) and L-glutamine (520 mg/L; Sigma, Louis, MO, USA) for up to 12 h.”
- Expand abbreviations (lncRNA, SPF, HOMA-IR).
I have added the full names of these abbreviations to the manuscript when they first appeared.
Line 49: long non-coding RNA (LncRNA)
Line 84: specific pathogen free (SPF)
Line 115: Homeostasis model assessment of insulin resistance (HOMA-IR)
- L.56: “LncRNA can sponge the miRNA” -> “LncRNA can sponge the microRNAs (miRNA)”
I have changed the sentence according to your advice.
Reviewer 2 Report
General comment:
The work of Feng and colleagues presents evidence of the benefits of methionine restriction on cognitive ability and exploits the underline mechanisms showing a relation between H19 overexpression due to a high-fat diet and ROS production and apoptosis.
Major issues:
1- This paper needs some English proofreading. Please sort the long sentences.
2- In Section 3.2 (lines 211-216), the authors refer that MR improved insulin signaling in HFD Mice by increasing IRS-1, GLUT1, HK2, PKM2, and PKF1, which were downregulated by HFD. In Figure 2, the expression of both PKM2 and PFK1 does not present significant differences in HFD and HFMR groups in comparison to the control group. Please correct these data and rephrase these sentences accordingly.
3- Several genes expression has been analyzed in this study. However, protein levels are not shown. Please provide complementary data by analyzing the correspondent protein expression.
4- Along the manuscript, it is demonstrated that MR can restore the levels of different genes as well as ROS levels to similar to those found in the Control group. This should be highlighted along the text.
5- The Section 5. Conclusions must be improved. It should briefly present the main conclusions of the study in an organized manner, and the contributions of these findings to research and clinical fields. Please review this section.
Minor issues:
1- In Figures 5 and 6, please correct the caption. Authors indicate “# (versus CON)” but in Figures 5A and 6A this is not accurate since the symbol “*” is used to show the statistical analysis.
2- In line 278, please rephrase the sentence. For example: “As shown in Figure 5, HG increased ROS content and H19 expression in a concentration-dependent manner, which was inhibited by MR treatment.”
3- In line 298, please rephrase the sentence.
4- In line 303, please rephrase the sentence. For example: “These results support our hypothesis that ROS production is related to H19 overexpression in hippocampal neurons.”
5- In Figure 8 and caption, please correct “victor” for “vector”.
Author Response
Dear professor,
Thank you for your valuable comments on this paper. I have revised the manuscript according to your suggestions. Thank you for reviewing my paper again. If there are any other suggestions, please let me know and I will revise them again.
Details of the revisions are as follows:
Major issues:
- This paper needs some English proofreading. Please sort the long sentences.
The English language editing of MDPI has checked and edited my manuscript.
- In Section 3.2 (lines 211-216), the authors refer that MR improved insulin signaling in HFD Mice by increasing IRS-1, GLUT1, HK2, PKM2, and PKF1, which were downregulated by HFD. In Figure 2, the expression of both PKM2 and PFK1 does not present significant differences in HFD and HFMR groups in comparison to the control group. Please correct these data and rephrase these sentences accordingly.
I have rephrased these sentences according to your advice.
“As shown in Figure 2, HFD reduced the glucose metabolism related mRNA expression of insulin receptor substrate 1 (IRS-1), glucose transporter 1 (GLUT1), hexokinase 2 (HK2) in the hippocampus, compared to the CON group (p < 0.05). However, these mRNA expressions were significantly increased in MR treated HFD mice. But, the mRNA ex-pression of M2 pyruvate kinase (PKM2) and phosphofructokinase 1 (PFK1) did not pre-sent significant differences in the HFD and HFMR groups in comparison to the CON group (p > 0.05).”
- Several genes expression has been analyzed in this study. However, protein levels are not shown. Please provide complementary data by analyzing the correspondent protein expression.
Thank you for your valuable suggestions. The results of the protein supplement will make this paper perfect. The direct regulatory targets of lncRNA and miRNA are downstream mRNA. Therefore, we mainly explored at the gene level and analyzed the correlation between H19 and caspase-3 mRNA expression. By inhibiting H19 expression, it was demonstrated that H19 can regulate caspase-3 mRNA expression via let-7. Therefore, we supplemented the protein expression level of caspase-3 to support our conclusion. In addition, it is difficult for us to add all protein results due to limited experimental conditions. We hope we can get your understanding.
Along the manuscript, it is demonstrated that MR can restore the levels of different genes as well as ROS levels to similar to those found in the Control group. This should be highlighted along the text.
I have revised my paper according to your suggestion.
Line 195-197: “Therefore, MR can restore the cognitive ability of the middle-aged mice fed with HFD to be similar to that of the CON group.”
Line 242-243: “As expected, MR can restore the levels of oxidative stress and ROS to similar to those found in the CON group.”
Line 312-313: “MR could significantly inhibit the increase of ROS and restore it to similar to those found in the CON group.”
Line 369-371: “In mice fed HFD, MR improved cognitive ability, increased the expression of learning and memory-related mRNA (BDNF, TRκB, CAMK2A, Synpo and CREB), improved insulin resistance, decreased blood glucose levels, and reduced oxidative stress, and restored these too similar to the CON group.”
- The Section 5. Conclusions must be improved. It should briefly present the main conclusions of the study in an organized manner, and the contributions of these findings to research and clinical fields. Please review this section.
I have rewritten the conclusion.
“In summary, MR can improve insulin resistance, alleviate oxidative stress, and inhibit oxidative stress-induced H19 overexpression in the hippocampus. Then, MR reduces hippocampal neuronal apoptosis through the H19/let-7/caspase-3 pathway (Figure 9), thereby improving the cognitive ability of middle-aged mice fed with HFD. This finding can provide a dietary reference for preventing or alleviating cognitive impairment in the elderly.”
Minor issues:
- In Figures 5 and 6, please correct the caption. Authors indicate “# (versus CON)” but in Figures 5A and 6A this is not accurate since the symbol “*” is used to show the statistical analysis.
I have corrected Figure 5A, Figure 6A, and Figure 6B.
- In line 278, please rephrase the sentence. For example: “As shown in Figure 5, HG increased ROS content and H19 expression in a concentration-dependent manner, which was inhibited by MR treatment.”
I have rephrased the sentence according to your advice.
- In line 298, please rephrase the sentence.
I have rephrased the sentence according to your advice.
“As shown in Figure 6AB, H2O2 increased ROS content and H19 expression in a con-centration-dependent manner, with a Pearson correlation coefficient between H19 and ROS of 0.8634 (Figure 6C).”
- In line 303, please rephrase the sentence. For example: “These results support our hypothesis that ROS production is related to H19 overexpression in hippocampal neurons.”
I have rephrased the sentence according to your advice.
- In Figure 8 and caption, please correct “victor” for “vector”.
I have corrected “victor” for “vector” in figure 8 and caption.